# ResSKNet-SSDP: Effective and Light End-To-End Architecture for Speaker Recognition

**DOI:** 10.3390/s23031203

**Published:** 2023-01-20

**Authors:** Fei Deng, Lihong Deng, Peifan Jiang, Gexiang Zhang, Qiang Yang

**Affiliations:** 1College of Computer Science and Cyber Security (Oxford Brookes College), Chengdu University of Technology, Chengdu 610059, China; 2Artificial Intelligence Research Center, Chengdu University of Technology, Chengdu 610059, China; 3School of Control Engineering, Chengdu University of Information Engineering, Chengdu 610059, China

**Keywords:** speaker recognition, end-to-end, selective kernel convolution, aggregation model

## Abstract

In speaker recognition tasks, convolutional neural network (CNN)-based approaches have shown significant success. Modeling the long-term contexts and efficiently aggregating the information are two challenges in speaker recognition, and they have a critical impact on system performance. Previous research has addressed these issues by introducing deeper, wider, and more complex network architectures and aggregation methods. However, it is difficult to significantly improve the performance with these approaches because they also have trouble fully utilizing global information, channel information, and time-frequency information. To address the above issues, we propose a lighter and more efficient CNN-based end-to-end speaker recognition architecture, ResSKNet-SSDP. ResSKNet-SSDP consists of a residual selective kernel network (ResSKNet) and self-attentive standard deviation pooling (SSDP). ResSKNet can capture long-term contexts, neighboring information, and global information, thus extracting a more informative frame-level. SSDP can capture short- and long-term changes in frame-level features, aggregating the variable-length frame-level features into fixed-length, more distinctive utterance-level features. Extensive comparison experiments were performed on two popular public speaker recognition datasets, Voxceleb and CN-Celeb, with current state-of-the-art speaker recognition systems and achieved the lowest EER/DCF of 2.33%/0.2298, 2.44%/0.2559, 4.10%/0.3502, and 12.28%/0.5051. Compared with the lightest x-vector, our designed ResSKNet-SSDP has 3.1 M fewer parameters and 31.6 ms less inference time, but 35.1% better performance. The results show that ResSKNet-SSDP significantly outperforms the current state-of-the-art speaker recognition architectures on all test sets and is an end-to-end architecture with fewer parameters and higher efficiency for applications in realistic situations. The ablation experiments further show that our proposed approaches also provide significant improvements over previous methods.

## 1. Introduction

Speaker recognition is intended to identify the speaker by obtaining identity information from the audio. With the widespread use of voice commands, speaker recognition has become a necessary measure to protect user security and privacy. However, the real-world recording environment can be too noisy for the audio to contain the speaker’s identity information. Intrinsic factors such as age, emotion, and intonation may also have an impact. Therefore, speaker recognition remains a challenging task. The key to achieving effective speaker recognition is to extract the fixed-dimensional and discriminative features from the audio.

In the past few decades, the traditional i-vector system with probabilistic linear discriminant analysis (PLDA) [1,2] was the principal method of speaker recognition. However, with the development of deep learning, deep neural networks (DNNs) have brought substantial improvements to speaker recognition. Compared with the traditional i-vector system, the DNN architecture does not require manual feature extraction. Instead, it can directly process noisy datasets to extract frame-level features and then aggregate the variable-length frame-level features into fixed-dimensional utterance-level features for end-to-end training. The DNN-based end-to-end speaker recognition system [3,4] achieved a better performance than the i-vector system and occupies a dominant position with excellent feature extraction capabilities. In recent years, researchers have been attempting to build more effective speaker recognition architectures to obtain more discriminative features. Specifically, these attempts can be divided into two categories: (1) more efficient network architectures; and (2) better aggregation models.

Convolutional neural networks (CNNs) are the most popular feature extractors for speaker recognition tasks. CNN-based [5,6] feature extractors are widely used due to their strong feature extraction capabilities. These feature extractors can be classified into two classes: (1) one-dimensional convolution-based structures [7,8,9,10,11,12]; and (2) two-dimensional convolution-based structures [13,14,15,16]. The one-dimensional convolution generates two-dimensional outputs with time and channel dimensions. The time-delay neural network (TDNN) [7,8,9] is a typical one-dimensional convolution structure. The two-dimensional convolutional structure treats the input acoustic features as an image with three dimensions—time, frequency, and channel—and uses two-dimensional convolution to produce three-dimensional outputs. In the two-dimensional convolutional structures, the time and frequency dimensions decrease, and the channel dimensions increase with the downsampling operation. ResNet is a representative two-dimensional convolutional structure [5]. However, CNNs have their imperfections. Convolutional operations use a fixed-size convolutional kernel to capture the time and frequency information of the audio. The size of the convolution kernel limits the receptive fields. As a result, the feature extractor is also limited in its capabilities, which leads to its inability to capture essential global information and model long-term contexts [17,18]. Some researchers have used deeper, wider, and more complex network structures to solve these problems. Although these methods can expand the receptive fields, they lead to a significant increase in parameters and inference time.

Another challenge is the aggregation of frame-level features. In speaker recognition, the length of the input audio is variable, and an aggregation model is needed to aggregate the variable-length features into fixed-dimensional utterance-level features after the feature extractor has acquired the frame-level features [19,20]. The most common approach is to use global average pooling (GAP) directly to aggregate frame-level features into utterance-level features. However, global average pooling causes the frame-level features to lose time and frequency information, and the aggregated utterance-level features are not discriminative. In addition, audio sometimes changes or pauses while speaking, and the global average pooling cannot focus on these significant parts. To address this problem, researchers have proposed attention-based aggregation models to aggregate the frame-level features [21,22,23]. However, these methods also use global average pooling on the frame-level features for pre-processing. They also lose some of the information and do not take full advantage of the audio’s time-frequency information.

On the other hand, with the widespread use of mobile devices [24,25,26,27], the design of speaker recognition systems tends to be light and efficient. However, existing models cannot be lighter, and the performance decreases drastically when made lighter. Therefore, we also need to design more light models for mobile devices.

To address these problems, we propose a more effective and lighter CNN-based end-to-end speaker recognition architecture, ResSKNet-SSDP. We propose a residual selective kernel network (ResSKNet), which can capture the neighborhood and global information of the features and can more efficiently model long-term contexts, resulting in more discriminative features. We propose self-attentive standard deviation pooling (SSDP). This avoids the pooling operation and preserves the time and frequency information, allowing a more accurate selection of information-rich frame-level features which contain more speaker identify information. Self-attentive standard deviation pooling can also capture the short- and long-term variations of the frame-level features, thus aggregating the variable-length frame-level features into fixed-length, robust, and more discriminative utterance-level features. To prove the effectiveness of our proposed method, we performed experiments with various settings in realistic environments. The main contributions of this study are as follows:We propose ResSKNet. This is a more efficient and lighter network structure that can capture global information and effectively model long-term contexts;We propose self-attentive standard deviation pooling (SSDP). This can capture short- and long-term changes in the frame-level features, aggregating the variable-length frame-level features into fixed-length, more distinctive utterance-level features;We build a light and efficient end-to-end speaker recognition system. Extensive experiments conducted on two popular public datasets demonstrate the effectiveness of the proposed architectures.

In the rest of this paper, we primarily present the most popular and advanced network structures and aggregation methods and their disadvantages in Section 2. In Section 3, we present the details of the proposed method. In Section 4, we introduce the dataset used, training details, testing details, and testing methods. In Section 5, we discuss and analyze the experimental results. Finally, we summarize the work conducted and explain the limitations of the current work and future research directions in Section 6.

## 2. Related Works

This section introduces the current research related to network structures and aggregation models for speaker recognition systems. In the network structure, we present the most common and state-of-the-art methods based on one-dimensional convolution and two-dimensional convolution at present. In the aggregation models, we describe the statistical-based approach, the attention-based approach, and the dictionary-based approach. In addition, we summarize the advantages and disadvantages of these methods.

### 2.1. Network Structure

To obtain a larger receptive field and improve the system performance, researchers usually improve the network structure, such as Snyder et al., who proposed a x-vector speaker recognition system [7]. It extracts frame-level features by overlaying TDNN layers and then uses statistical pooling to aggregate the frame-level features into a fixed-dimensional speaker vector. The x-vector has become the most commonly used method for speaker recognition due to its excellent performance and light structure. The structural design of the x-vector also laid the foundation for most of the speaker recognition systems, such as E-TDNN [7], F-TDNN [8], DTDNN [18], and ECAPA-TDNN [12]. In addition, many works have used famous CNN structures, such as ResNet [5], ResNeXt, and Res2Net [16], in extracting the frame-level features. ResNet [5] introduced short connections to the neural network, thus alleviating the problem of gradient disappearance, and obtaining a deeper network structure, simultaneously. ResNet is also the most popular feature extractor in current speaker recognition systems. ResNeXt [16] adopts the repetitive layer strategy of VGG/ResNets while exploiting the split-transform-merge strategy in a simple and scalable way. A module in ResNeXt performs a set of transformations, each on a low-dimensional feature, whose output is aggregated by summation. ResNeXt reduces the parameters and inference time of the model, but it has yet to improve its modelling of long-term contexts. Res2Net [16] focuses on the revision of ResNet blocks, which build hierarchical residual block-like connections in a single residual block. It splits the features within one block into multiple channel groups and designs residual-like connections across the different channel groups. This residual-style connection increases the coverage of the receptive fields and yields many different combinations of the receptive fields for the improved modeling of long-term contexts. Res2Net has a smaller parameter, while achieving a better performance improvement. Although these methods achieved better performance improvement, they require more computation resources and are difficult to lighten the structure. The light structure makes it difficult to capture global information and model long-term contexts. Therefore, we designed ResSKNet. It combines the advantages of both regular and dilation convolution and uses an attention mechanism to adjust the weights between the two convolutions according to the input features. Thus, it can capture global information and adaptively adjust the weights between the short-term and long-term contexts of the input audio to better model long-term contexts.

### 2.2. Aggregation Model

In speaker recognition, the length of the input acoustic features is variable. Therefore, a flexible processing method should have the ability to accept audio of any duration and obtain the fixed-dimensional features. In speaker recognition systems, the GAP aggregation model [10,20,24] is the most commonly used method for aggregating the frame-level features into fixed-dimensional utterance-level features. The reference [7] employs statistical pooling (SP) to aggregate the features, which computes the mean vector of the frame-level features and the standard deviation vector of the second-order statistics; then, it stitches them together as an utterance-level feature. However, our voices sometimes change, and there are brief pauses in the audio. Hence, researchers have proposed a self-attentive pooling (SAP) aggregation model [3,21,22] based on the attention mechanism to solve this problem and select frames that contain speaker information more effectively. Okabe et al. constructed the attention statistics pooling (ASP) aggregation model [23] based on the SP aggregation model by introducing an attention mechanism. It is calculated in the same way as the SP aggregation model, but has better performance than SP. They also introduced the NetVLAD aggregation model in computer vision that aggregates features into fixed dimensions via clustering [20,28]. NetVLAD assigns each frame-level feature to a different cluster center and encodes the residuals as output features. However, attention-based pooling methods are weakly robust and can exhibit a lower performance than GAP in different experimental settings. In addition, these methods usually use pooling operations to process features, resulting in a loss of information in the time and frequency dimensions and weakening the discriminative character of utterance-level features. Chung et al. conducted experiments using the NetVLAD method and its variants on the Voxceleb dataset and achieved the best results for speaker recognition [21]. It is a more robust and effective aggregation model, but the number of parameters increases as the cluster center increases. Therefore, we propose self-attentive standard deviation pooling. We avoid the pooling operation and retain the time and frequency information of the features, which allows self-attentive standard deviation pooling to select information-rich frames and capture the short- and long-term changes of the frame-level features more accurately. Thus, it aggregates more discriminative utterance-level features. In addition, it has stronger robustness and fewer parameters.

## 3. Materials and Methods

In this section, we introduce the proposed architecture, ResSKNet-SSDP, as shown in Figure 1. It consists of four parts: (1) feature extraction network. We use ResSKNet to extract frame-level features, which can acquire local and global information more efficiently to model both short-term and long-term contexts, as shown in the red box in Figure 1. The structure of ResSKNet is shown in Table 1; (2) feature aggregation. The aggregation model is a bridge between the frame-level features and the utterance-level features. We aggregate the variable-length frame-level features into the fixed-dimensional utterance-level features through self-attentive standard deviation pooling, as shown in the blue box in Figure 1; (3) speaker recognition loss function. AM-softmax loss is used during training to classify speakers more accurately [29]; (4) similarity metric. This is used to identify speakers, after the training, by calculating the distance of a pair of utterance-level features to determine whether the audio comes from the same speaker.

### 3.1. ResSKNet

Dilated convolution is a special convolutional operation that skips the input values by a certain step and uses filters in the region over its convolution kernel. Compared to standard convolution, dilated convolution can produce a larger receptive field by skipping and can more effectively model variable-length audio with long-term contexts. However, it is a challenge to use dilation convolution in deep convolutional neural networks. Performing lots of dilated convolutional layers in a deep structure will lead to a gridding problem that causes the complete loss of the local information and makes the information at a distance unrelated. In addition, it is problematic to determine in which layers to use dilation convolution for deep structures.

To solve the above problem, we designed a ResSKNet block, which consists of a residual selective kernel convolution (ResSKConv) module and 1 × 1 convolution with residual connections. We improved and introduced the selective kernel convolution (SKConv) module [18] into the end-to-end speaker recognition architecture, as shown in Figure 2. It performs standard and dilated convolutional layers on two parallel paths to capture the local and global information to better model short- and long-term contexts. Then, we use a self-attentive module to obtain the global information and adaptively adjust the weight between the short-term and long-term contexts.

Using *x*∈*R^T^*^×*F*×*C*^ to denote the input features, *T*, *F*, and *C* are the time dimensions, frequency dimensions, and channel dimensions, respectively. SKConv performs 3 × 3 convolutional *F*′ and 3 × 3 dilated convolutional *F*″, respectively. The dilation rate is set to 2.
(1)$$u=\delta (B(F^{\prime }(x)))$$
(2)$$u^{\prime }=\delta (B(F^{\prime \prime }(x)))$$

In Equations (1) and (2), *B* and *δ* denote the batch normalization (BN) and ReLU activation functions, respectively. We do not use addition to fuse the features on the two parallel paths. Instead, we connect the features of the two branch outputs along the channel dimension and use a 1 × 1 convolution to fuse them, where *U*∈*R^T^*^×*F*×2*C*^ is the global feature obtained from the two paths. Compared to the addition method, we use convolution to fuse the features of two parallel paths along the channel dimension, thus increasing the depth and making full use of the features. Then, we generate a global channel feature *U_C_*∈*R*^1×1×*C*^ using global average pooling and perform a feature transformation on it to generate channel attention *w_CF_*∈*R*^1×1×*C*^ and *w_CF_*_′_∈*R*^1×1×*C*^ on the two paths. Finally, after connecting the weighted elements of the features on both paths along the channel dimension, the features *y*∈*R^T^*^×*F*×*C*^ are then output by using a 1 × 1 convolution.
(3)$$U=conv_{1}(u,u^{\prime })$$
(4)$$U_{C}=GAP(U)$$
(5)$$w_{CF}=\mathrm{softmax}(conv_{3}(\delta (conv_{2}(U_{C}))))$$
(6)$$w_{CF^{\prime }}=\mathrm{softmax}(conv_{4}(\delta (conv_{2}(U_{C}))))$$
(7)$$y=conv_{5}(u\times w_{CF},u^{\prime }\times w_{CF^{\prime }})$$
where *conv*_1_∈*R*^1×1×*C*^, *conv*_2_∈*R*^1×1×^*^C/r^*, *conv*_3_∈*R*^1×1×*C*^, *conv*_4_∈*R*^1×1×*C*^, and *conv*_5_∈*R*^1×1×*C*^ are the convolution operations, *r* is the scale factor used to reduce the parameters and to obtain the dependencies between channels. Usually, we set *r* to 16. We do not use a fully connected layer for feature transformation because a fully connected layer is inefficient and useless. We use a 1 × 1 convolution, which captures the dependencies between channels more efficiently and thus produces more accurate channel attention [30].

Current speaker recognition systems only use the features of the last convolutional layer to obtain the frame-level features. Considering the hierarchical character of CNNs, these deeper features are the most complex and should be closely related to the speaker’s identity. However, according to references [31,32], we believe that shallow features are also helpful in making utterance-level features more discriminative. Therefore, in our proposed system, we connect the output features of each stage of the ResSKNet block. We use 1 × 1 convolution to transform the output of each stage to the same size as the output of the last stage. The final architecture is shown in Figure 1.

### 3.2. Self-Attentive Standard Deviation Pooling

Attention-based aggregation models, such as SAP and ASP, can select frame-level features based on their importance. However, they can exhibit a similar or lower performance than GAP in different experimental settings. This indicates that these attention-based aggregation models are not accurate in selecting more informative frame-level features and have weak robustness. In addition, these methods usually use pooling operations to process features, resulting in the loss of time and frequency information and reducing the utterance-level feature differentiation. Therefore, we propose self-attentive standard deviation pooling. Self-attentive standard deviation pooling avoids pooling operations and preserves the time and frequency information of the frame-level features, allowing a more accurate selection of information-rich frames. At the same time, the self-attentive standard deviation pooling computation can capture the short- and long-term changes of frame-level features, aggregating variable-length frame-level features into fixed-length, more differentiated utterance-level features. It is also more robust than the previous attention-based aggregation models.

Use *x*∈*R^T^*^×^*^F^*^×*C*^ to denote the frame-level features extracted by the convolutional neural network (*T* is the time dimension, *F* is the frequency dimension, and *C* is the channel dimension). The existing aggregation methods typically use an average pooling layer along the frequency axis of the features to generate a time feature descriptor matrix *X*∈*R^T^*^×*C*^ and *h* = [*x*_1_,*x*_2_,…,*x_T_*], where *x_t_*∈*R*^1×*C*^. However, this causes the features to lose information in the frequency dimension and limits the performance of the aggregation model. We transform *x* to obtain the time-frequency feature description matrix *H*∈*R^N^*^×*C*^ (*N* = *T* × *F*) and *h* = [*h*_1_,*h*_2_,…,*h_N_*], where *h_t_*∈*R*^1×*C*^. We retain the time and frequency information of the features, and then use it to generate the importance weight *w_t_*∈*R*^1×*C*^ corresponding to each frame-level feature. Considering that the frame-level features extracted by the neural network already have great screening, the non-linear activation function is not applied to change the distribution of the features during aggregation, causing feature distortion. Therefore, we use a linear attention mechanism with a softmax function in the self-attentive standard deviation aggregation model to generate the weights, as shown in Equation (8). The *f_SL_*() denotes the linear attention mechanism and it can enhance the significant parts of the frame-level features and suppress the unessential parts without causing feature distortion. The non-linear activation functions change the information in the frame-level features and the feature distribution, resulting in feature distortion and the inaccurate selection of frame-level features. Thus, self-attentive standard deviation pooling retains more speaker information and can select frame-level features more accurately.
(8)$$e_{t}=f_{SL}\left(h_{t}\right)=w^{T}h_{t}+b$$
(9)$$w_{t}=\frac{exp\left(e_{t}\right)}{\sum_{i=1}^{T}exp\left(e_{i}\right)}$$

The mean vector *μ*∈*R*^1×*C*^ and the standard deviation vector *σ*∈*R*^1×*C*^ are calculated as shown in Equations (10) and (11). However, statistical methods cannot identify which part of the audio contains the speaker identity information. Therefore, we use attention to calculate the mean vector and the standard deviation vector.
(10)$$\mu =\frac{1}{T}\sum_{t}^{T}h_{t}$$
(11)$$\sigma =\sqrt{\frac{1}{T}\sum_{t}^{T}h_{t}⊙h_{t}-\mu ⊙\mu }$$
where *h_t_* denotes the time-frequency feature descriptions, and ⊙ denotes the Adama product.

In self-attentive standard deviation pooling, we obtain the mean vector by weighted summation, as shown in Equation (12): *w_t_* is the importance weight, and *α*∈*R*^1×*C*^ is a learnable vector. The average vector generated by attention can be trained along with the neural network. It is learnable. In addition, it can suppress noise, thus filtering out part of the interference information to retain more effective time-frequency information. Finally, the features *h_t_* and weights *w_t_* are concatenated and reduced with the mean vector *μ* to obtain the self-attentive standard deviation vector *e*∈*R^C^*, as shown in Equation (13). The self-attentive standard deviation vector is the utterance-level features obtained from the aggregating frame-level features. As the standard deviation contains other speaker features, in terms of time variability over long-term contexts, the self-attentive standard pooling model captures features’ long-term variation.
(12)$$\mu =\sum_{t=1}^{N}w_{t}\alpha$$
(13)$$e=\frac{\sum_{t=1}^{N}w_{t}h_{t}-\mu }{N}$$

We did not use the same calculation as the statistical method because the frame-level features obtained by the convolutional neural network are first-order information, and there is an information mismatch between them and the higher-order information obtained by the statistical method [33]. In contrast, self-attentive standard deviation pooling uses first-order information for calculation throughout the process, and the output utterance-level features are also first-order information, avoiding this mismatch. At the same time, in the audio features, the difference in frequencies between different time-frequency locations is very large, and there is a coincidence in the direct addition of the results. The self-attentive standard deviation calculation eliminates this chance and makes the produced utterance-level features more discriminative. It goes without saying that self-attentive standard deviation pooling is also differentiable. Therefore, self-attentive standard deviation pooling can also be trained, along with the speaker recognition system, based on backpropagation.

## 4. Experimental Setups

### 4.1. Dataset

The experimental speaker datasets were adopted from the CN-Celeb [34] and Voxceleb [19] datasets, which have been commonly used for speaker recognition tasks in recent years.

There are 3000 speakers and 11 different genres in CN-Celeb, which include various real-world noises, cross-channel mismatches, and speaking styles in wild speech utterances. The training set has more than 600,000 utterances from 2800 speakers and 18,024 utterances from 200 speakers. There are 3,604,800 pairs in the test trials. Moreover, domain mismatch between the enrollment and test in the trials makes this dataset a very challenging one in speaker verification.

Voxceleb is a large text-independent speaker recognition dataset containing the Voxceleb1 and the Voxceleb2 dataset, and Voxceleb2 contains more than 1 million audio clips of 5994 speakers extracted from YouTube videos. The average time duration was 7.8 s, from different acoustic environments, making speaker identification more challenging. Voxceleb1 contains over 100,000 pieces of audio from 1251 speakers. There are three test sets: Voxceleb1-O, Voxceleb1-E, and Voxceleb1-H. Voxceleb1-O is a test set that includes 40 speakers independent of Voxceleb1 and does not overlap with the speakers in Voxceleb1. The Voxceleb1-E test set uses the entire Voxceleb1 dataset, while the Voxceleb1-H test set is specific. It contains samples from the same country of nationality and the same gender. The Voxceleb2 dataset is an extended version of the Voxceleb1 dataset, but the two datasets are mutually exclusive. As mentioned in reference [31], Voxceleb2 contains some flaws in its annotation. Therefore, it is not recommended for testing models. It is widely used for training. As with most existing references, we use Voxceleb2 for training and Voxceleb1 as the test set.

### 4.2. Training Details

We selected the 40-dimensional Filter Banks as the input features to the deep convolutional neural network without voice activity detection (VAD) and data augmentation [35]. The acoustic features have a frame length of 25 ms and a frame shift of 10 ms. Compared with the other acoustic features [36,37], Filter Banks are more in line with the nature of the sound signal and suitable for the reception characteristics of the human ear. We also do not use complex processing at the back-end, such as PLDA. During the training, we cut out a three-second clip from the audio.

The loss function of the system adopts AM-softmax [29] with margin = 0.1 and scale = 30. Compared with the softmax loss function, AM-softmax improves the verification accuracy by introducing a boundary in the angular space. It is calculated as follows:(14)$$L_{i}=-\log\frac{e^{s(\cos\theta_{yi}-m)}}{e^{s(\cos\theta_{yi}-m)}+\sum_{j\neq y_{i}}e^{s\cos(\theta_{j})}}$$
where *L_i_* is the cost of classifying the sample correctly, and *θ_y_* = *arccos* (*w*^*T*^*x*) refers to the angle between the sample feature and the decision hyperplane (*w*), with both vectors normalized by L2. Therefore, the angle is minimized by making cos(*θ_yi_*)−*m* as large as possible, where m is the angle boundary. The hyperparameter s controls the “temperature” of the loss function, producing higher gradients for well-separated samples and further reducing the intra-class variance.

The Adam optimizer [38] with an initial learning rate of 0.001 was used to optimize the network parameters and attenuate 0.1 times every five cycles for training.

### 4.3. Testing and Testing Standards

During the test phase, we used the same settings as [19], extracted ten 3-s segments from each test audio as samples, and then sent them to the system to extract the utterance-level features of each segment and to calculate the distance between all of the combinations (10 × 10 = 100) of pairs of segments. Then, the average of 100 distances denotes the score.

This study adopts the commonly used equal error rate (EER) and minimum detection cost function 2010 (DCF10) [39] as the evaluation indices to objectively evaluate the performance of different aggregation models. They indicate that the smaller the value, the better the performance. The calculation formula of the minimum detection cost function is:(15)$$DCF=C_{FR}F_{FR}P_{target}+C_{FA}F_{FA}(1-P_{target})$$
where *C_FR_* and *C_FA_* are the weights of false rejection rate *F_FR_* and false acceptance rate *F_FA_*, respectively, and *P_target_* and 1−*P_target_* are the prior probability of real speaking and impersonation tests. We use the parameters *C_FR_* = 1, *C_FA_* = 1, and *P_target_* = 0.01 (DCF10) set by NIST SRE2010. DCF not only considers the different costs of false rejection and false reception but also considers the prior probability of the test. Therefore, MinDCF is more informative than EER in model performance evaluation.

## 5. Results

### 5.1. Ablation Experiments

#### 5.1.1. Evaluating the Residuals Selective Kernel Convolution

To validate the effectiveness of the proposed ResSKNet-SSDP speaker recognition system, we first performed a series of ablation experiments. In addition, we counted their parameters and inference time. The inference time is the time required by the speaker recognition system to convert the audio into the utterance-level features. We both used the same experimental setups, simple training methods, and direct aggregation of frame-level features using global average pooling (GAP). Table 2 shows the test results of the regular convolution, SK conv, with our proposed ResSK conv on voxceleb1-O. As shown in Table 2, the EER/DCF of the regular convolution is 4.17%/0.3882. After using SK conv, the EER/DCF decreases to 3.42%/0.3522. It significantly improves the system performance, although the parameters and inference time increased slightly. It also shows that using SK conv improves the modelling of the long-term contexts. Testing again using our proposed ResSK conv as shown in Table 2, the EER/DCF further decreased to 2.85%/0.3126. This indicates that our ResSK conv obtained a larger perceptual field than the SK conv by using dilated convolution. In addition, we captured the dependencies between channels more effectively by using 1 × 1 convolution in the ResSK conv than the fully connected layer in the SK conv, which produces more accurate channel attention. These improvements allow ResSK conv to capture local and global information more effectively and thus better model short- and long-term contexts. Compared to the regular conv, it increases the parameters by 0.3 M and inference time by 4.1 ms, but the performance is improved by 31.7%. Compared with SK conv, it increases the inference time by 1.7 ms with 0.1 M more parameters, but the performance is improved by 16.7%.

#### 5.1.2. Evaluating the Dilation Rate of the Residuals Selective Kernel Convolution

As shown in Table 3, when the dilation rate is 0, the EER/DCF is 3.29%/0.3417, and the perceptual field of the convolution at this time is consistent with that of the conventional convolution. When the dilation rate is 1, the EER/DCF decreases to 3.16%/0.3253. At this time, the ResSK conv has a larger receptive field. When the dilation rate is 2, the EER/DCF further decreases to 3.16%/0.3253, the receptive field also increases further and achieves the best performance. However, as the dilation rate increases, the performance of ResSK conv gradually decreases. The increase in the dilation rate leads to the loss of a lot of local information and makes the information discontinuous and irrelevant. In addition, we found that the inference time gradually increased with an increase in the dilation rate. This is because, as the dilation rate increases, the convolutional kernel of ResSK conv also gradually increases, and it also limits the bottleneck of the inference speed due to the GPU access memory bandwidth. Therefore, in this study, the best performance of ResSK conv is achieved when the dilation rate is 2, which can capture local and global information more effectively and thus better model the long-term contexts.

#### 5.1.3. Evaluating the ResSKNet

Table 4 shows the results of ResSKNet with other networks on the Voxceleb1-O test, set under the same experimental conditions. As shown in Table 4, the worst performers were x-vector [7] and ResNet-34 [19], with EER/DCF of 4.39%/0.3726 and 4.47%/0.3909, respectively. However, compared to ResNet-34, the x-vector has the lightest structure and shorter inference time with fewer parameters. Due to its deeper and wider network structure, ResNet-50 [19] exhibits a significant performance improvement, with the EER/DCF falling to 3.89%/0.3710. However, both the inference time and its parameters increase dramatically. We then tested with ResNeXt [16] and found that ResNeXt achieved a similar performance to ResNet-50 with an EER/DCF of 3.85%/0.3822. Compared to ResNet-50, ResNeXt has 12.2 M fewer parameters and 55.2 ms less inference time. This indicates that the structure of ResNeXt can effectively reduce the parameters and inference time of the model, but it still cannot better model the long-term context. Testing Res2Net [16] again, we found that Res2Net achieves the lowest EER/DCF, of 3.32%/0.3442, compared to the previously tested methods. Its parameters and inference time are close to ResNeXt but still much higher than the x-vector structure. This suggests that Res2Net is a more efficient network structure that can better model long-term contexts, but is still not applicable in practice due to its parameters and inference time. Finally, we tested the proposed ResSKNet. ResSKNet further improves its performance compared to Res2Net, achieving the lowest EER/DCF of 2.85%/0.3126. This indicates that ResSKNet has better feature extraction capability and can more effectively model long-term contexts and our designed ResSKNet structure makes full use of both shallow and deep features, resulting in more informative frame-level features with more speaker identity information. Meanwhile, it has 2.1 M fewer parameters and 36.1 ms less inference time compared to the x-vector, but a 35.1% improvement in performance. This proves that the ResSKNet we constructed has a more efficient and light structure. To further visualize the effectiveness of the proposed architecture, we plotted detection error trade-off (DET) curves for all comparable models, as shown in Figure 3. We found that ResSKNet maintained a great performance advantage and was always below all of the curves.

#### 5.1.4. Evaluating the SSDP Aggregation Model

Table 5 shows the performance of our proposed self-attentive standard deviation pooling with other aggregation models on the Voxceleb1-O test set. Similarly, both the training and testing were performed under the same experimental conditions, and we did not introduce the attention method into the network. We directly use ResSKNet as the backbone network. As shown in Table 5, the performance of the baseline system can also be significantly improved using the aggregation model. The SP aggregation model [7] reduces the EER/DCF of the system to 2.64%/0.3883 by calculating the mean and standard deviation vector of the frame-level features. The SAP aggregation model [21] further improves its performance by focusing on more significant frame-level features, achieving a lower EER/DCF than the statistical approach of 2.62%/0.3599. The ASP aggregation model [23] combines attention and statistics to further reduce the EER/DCF to 2.57%/0.3563. However, it is not a significant improvement over the SAP aggregation model. This indicates that the ASP aggregation model does not effectively combine attentional and statistical methods. NetVLAD [28] still shows the best performance with an EER/DCF of 2.42%/0.3391. Finally, we tested the SSDP proposed in this paper. We found that SSDP achieves the lowest EER/DCF of 2.33%/0.2451, which is better than NetVLAD. This proves that the utterance-level features obtained from SSDP are more discriminative. It can more accurately select frame-level features and capture short- and long-term changes in the frame-level features, resulting in robust and more discriminative utterance-level features. Compared to ASP, it combines attention and statistics methods more effectively. We estimated the parameters and inference time of these aggregation models. The proposed SSDP has 0.2 M more parameters and 4.4 ms more inference time than the GAP, with the simplest structure and the least number of parameters, but the performance is improved by 18%. On the other hand, compared to NetVLAD, self-attentive standard deviation pooling has fewer parameters, a smaller inference time, and better performance. We similarly plotted the detection error trade-off (DET) curves for all of the comparable aggregation models, as shown in Figure 4. Our proposed SSDP also significantly outperforms the NetVLAD, which had previously achieved the best results, and is always below the NetVLAD curve.

### 5.2. Results of the Speaker Recognition Systems on Voxceleb1-O

Next, our proposed ResSKNet-SSDP end-to-end speaker recognition system is compared and evaluated on the Voxceleb1-O test set with these current advanced speaker recognition systems, as shown in Table 6. As some of the methods are not open source, we directly applied the results from the paper. In the previous work, RawNet3 [40] and ECAPA-TDNN [12] were the most advanced methods in speaker recognition systems with EER/DCF of 2.35%/0.2513 and 2.38%/0.2349. However, our ResSKNet-SSDP end-to-end speaker recognition system outperforms the previous best results, achieving the lowest EER/DCF of 2.33%/0.2298. This proves that the ResSKNet-SSDP end-to-end speaker recognition system has a greater feature extraction ability and is a lightweight speaker recognition system suitable for practical applications.

To evaluate the performance of the ResSKNet-SSDP end-to-end speaker recognition system more comprehensively, we tested it again in the more extensive and challenging Voxceleb1-E and Voxceleb1-H test sets, as shown in Table 7. In Voxceleb1-E, which uses the entire Voxceleb1 as the test set, the proposed ResSKNet-SSDP end-to-end speaker recognition system still achieves the best results with an EER/DCF of 2.44%/0.2559. In the Voxceleb1-H test set, using the same country and gender, the differences in accent and intonation decreased, and they were more difficult to distinguish. As a result, the EER/DCF increased for all systems. However, our proposed ResSKNet-SSDP end-to-end speaker recognition system still holds the lead with an EER/DCF of 4.10%/0.3502 below other methods. This demonstrates that the ResSKNet-SSDP end-to-end speaker recognition system can obtain more distinguishable features and thus better distinguish between speakers with higher similarity. In the three test sets of Voxceleb, our designed ResSKNet-SSDP also maintains a great advantage, and the DET curve is always below the other systems’ curves, as shown in Figure 5.

### 5.3. Results of the Speaker Recognition Systems on CN-Celeb

Table 8 shows the test results of these speaker recognition systems on CN-Celeb. It can be seen that the results on CN-Celeb are worse because the registered and tested discourse is shorter in this dataset. For example, the EER/DCF on CN-Celeb is substantially and obviously increased compared to the x-vector results for Voxceleb1-O. These results clearly show that state-of-the-art speaker recognition systems cannot inherently deal with the complexity introduced by multiple genres. In previous work, EDINet achieved the best identification results on CN-Celeb, with an EER of 12.8%. The ResSKNet-SSDP end-to-end speaker recognition system we have designed achieves slightly better performance than EDINet, with an EER/DCF of 12.81%/0.5051. In addition, we can clearly see that the DET curves of all of the systems have shifted significantly upwards compared to the Voxceleb dataset, but the curve of ResSKNet-SSDP is still lower than the other systems, as shown in Figure 6. The ResSKNet-SSDP end-to-end speaker recognition system achieved an excellent performance on all of the different test sets, proving that it is more efficient, lighter, and more suitable for practical application.

### 5.4. Visualization Analysis

We used the visualization method by Kye S M et al. to visualize the effectiveness of the proposed ResSKNet-SSDP end-to-end speaker identification system. We formed the visualization map after the dimensionality reduction in the speaker identity features by the t-SNE [46]. In Voxceleb1-H, fifty speakers were randomly selected to be represented by different colors. Each person randomly selected ten audios, and then ten randomly extracted three-second test segments from each audio. There were a total of 5000 three-second test segments obtained. The visualization maps of x-vector, Res2Net, ResSKNet, and ResSKNet-SSDP are shown in Figure 7.

(a) shows the speaker feature visualization graph of x-vector. It is observed that the feature extraction ability of x-vector is weak. The result is that the visualization graph is also very poorly classified, with many classification errors and collisions occurring. This suggests that the speaker features obtained by x-vector are not discriminative. (b) shows the visualization of Res2Net. It is noticeable that its visualization has been significantly improved compared to (a). It has fewer classification errors and collisions. It verifies the effectiveness and robustness of Res2Net. It also shows that Res2Net has a better feature extraction ability, and the speaker features are more discriminative. However, the intra-class distance of the speaker features is bigger, and the inter-class is smaller, and there are still some classification errors and collisions, as shown in (b) in Figure 7c shows the visualization map of ResSKNet. Comparing the visualization map of Res2Net, we can see that the collision in the visualization map of ResSKNet is significantly improved, and almost no speaker features collide. The inter-class distance increases significantly. It also proves that ResSKNet effectively improves the feature extraction ability by acquiring larger receptive fields and generating more combinations of receptive fields. However, there are still a few cases of classification errors and large intra-class distances in ResSKNet, as shown in (c) in Figure 7d represents the visualization map of ResSKNet-SSDP. It not only has no classification errors, but also has a larger inter-class distance and closer intra-class distance. This indicates that our proposed SSDP is efficient. It effectively improves the feature extraction ability of ResSKNet-SSDP to achieve more distinguishable features and it gives the speaker features from the same speaker a higher similarity.

## 6. Conclusions

In this work, we designed a speaker recognition architecture, ResSKNet-SSDP, with an improved feature extraction capability and improved adaptation to the speaker recognition task. The ResSKNet-SSDP network models the long-term contexts more effectively through the ResSKNet network structure. In addition, it also introduces the SSDP aggregation model to capture the short- and long-term changes of the frame-level features, aggregating variable-length frame-level features into fixed-length, more differentiated utterance-level features. We achieved the lowest EER/DCF of 2.33%/0.2298, 2.44%/0.2559, 4.10%/0.3502, and 12.28%/0.5051 on the noisy Voxceleb1 and CN-Celeb test sets, outperforming many of the existing methods and effectively improving the accuracy of the end-to-end speaker recognition system. Our proposed ResSKNet-SSDP end-to-end speaker recognition system has 3.1 M fewer parameters and 31.6 ms less inference time compared to the lightest x-vector, but 35.1% better performance, which indicates that it is a more efficient and lightweight structure. It is also more suitable for practical application.

The work we have conducted has improved the feature extraction ability of the speaker recognition system. When performing speaker recognition, we need to map the utterance-level features into the features space for similarity metrics. Therefore, the essence of speaker recognition is a metric learning problem and how to make the network more effective for metric learning is also a critical issue. The loss function is the key to solving this problem. In future work, we will focus on improving the loss function. In addition, we will also extend the proposed approach to other voice applications, such as language recognition and emotion recognition.

## Figures and Tables

**Figure 1 sensors-23-01203-f001:**
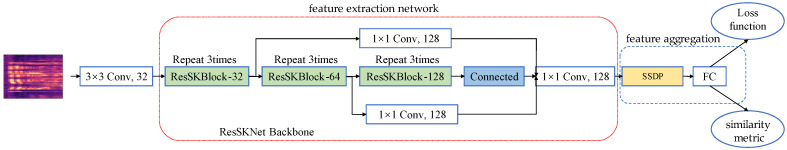
Overview of the ResSKNet-SSDP speaker recognition system.

**Figure 2 sensors-23-01203-f002:**
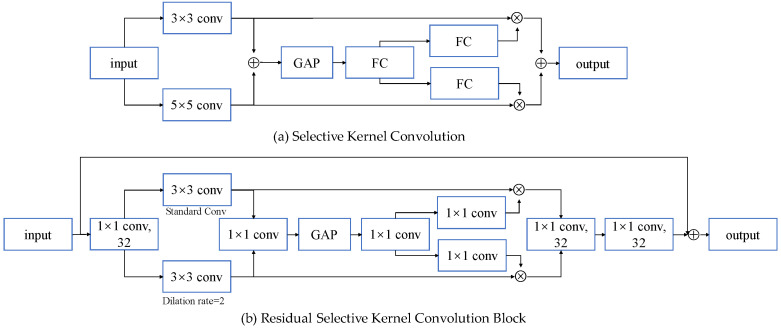
Residual selective kernel convolution structure.

**Figure 3 sensors-23-01203-f003:**
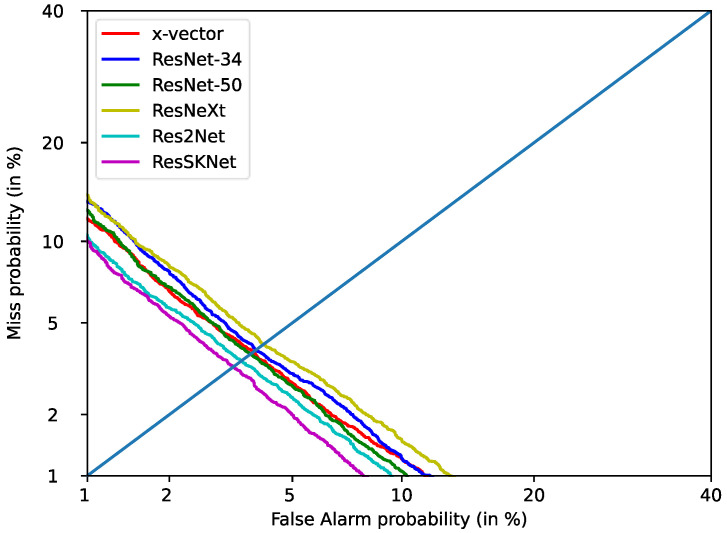
DET curve of different networks on Voxceleb1-O.

**Figure 4 sensors-23-01203-f004:**
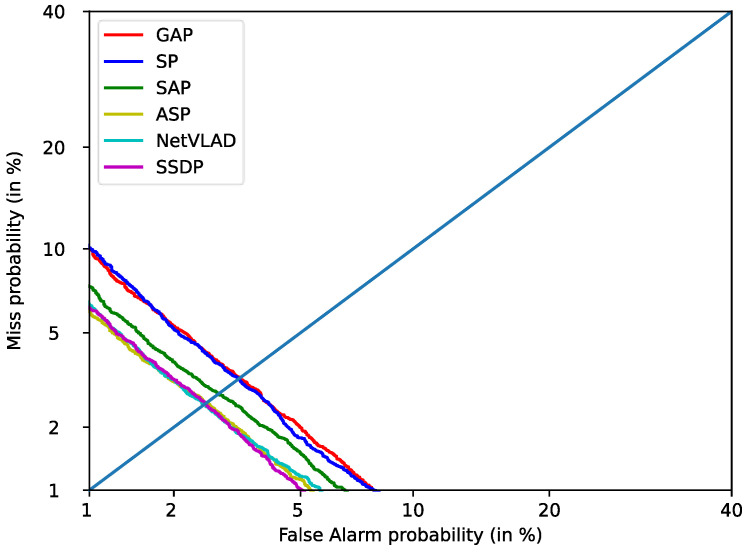
DET curve of aggregation models on Voxceleb1-O.

**Figure 5 sensors-23-01203-f005:**
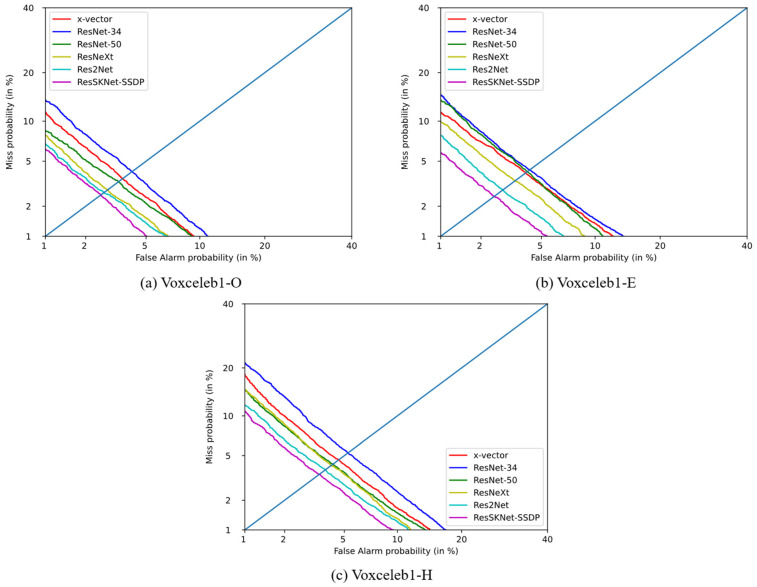
DET curve of different systems on Voxceleb1 test set.

**Figure 6 sensors-23-01203-f006:**
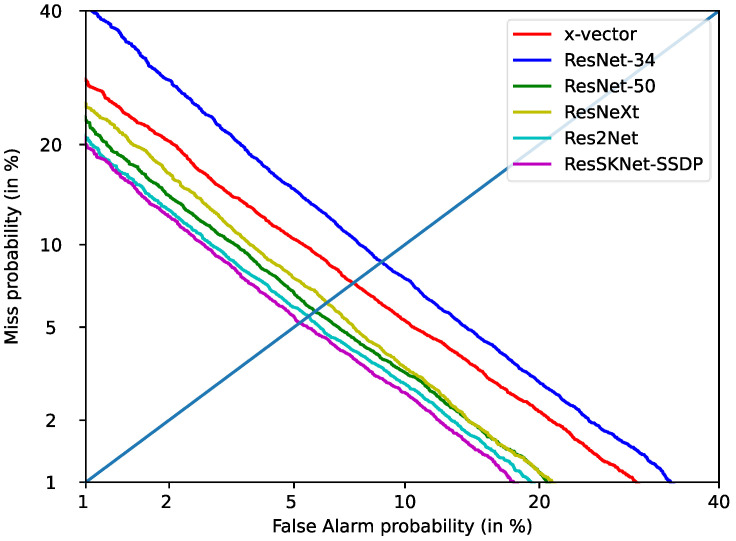
DET curve of systems on CN-celeb.

**Figure 7 sensors-23-01203-f007:**
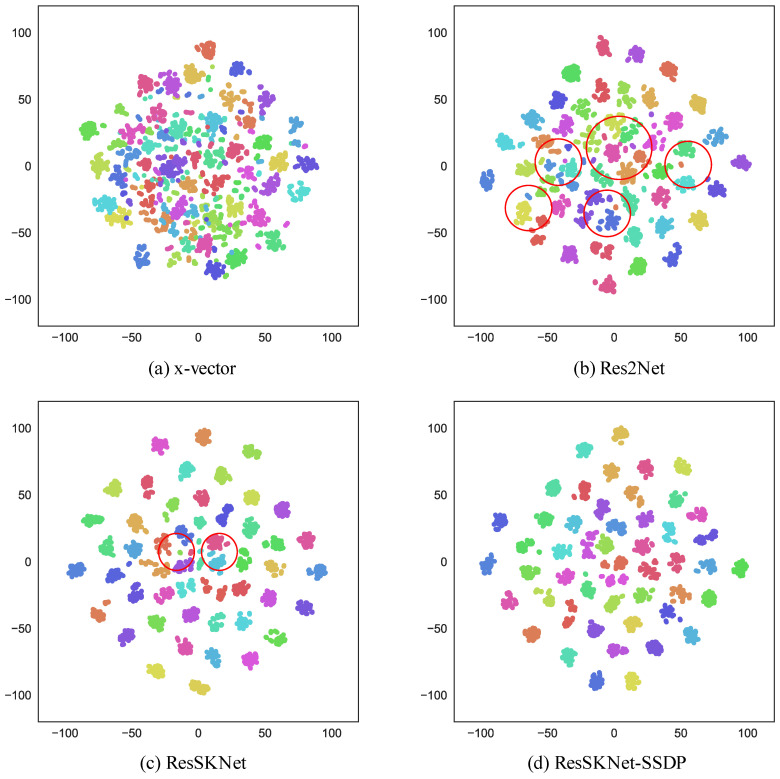
t-SNE visualization results.

**Table 1 sensors-23-01203-t001:** The structure of ResSKNet.

Layer	Blocks (*T* × 40 × 1)	Channels	Output Size (*T* × *F* × *C*)
Conv 1	Conv2d, 3 × 3, stride 1	32	*T* × 40 × 16
Stage 1	ResSK Block×3, stride 2	32	*T* × 40 × 32
Stage 2	ResSK Block×3, stride 2	64	*T*/2 × 20 × 66
Stage 3	ResSK Block×3, stride 2	128	*T*/4 × 10 × 128

**Table 2 sensors-23-01203-t002:** Results of residual selective kernel convolution on Voxceleb1-O.

Model	Method	Aggregation	Loss	Dims	EER (%)	DCF	Params (M)	Time (ms)
ResSKNet	regular conv	GAP	AM-softmax	512	4.17	0.3882	2.9	27.8
ResSKNet	SK conv	GAP	AM-softmax	512	3.42	0.3522	3.1	30.2
ResSKNet	ResSK conv	GAP	AM-softmax	512	2.85	0.3126	3.2	31.9

**Table 3 sensors-23-01203-t003:** Results of dilation rate on Voxceleb1-O.

Model	Dilation Rate	Aggregation	Loss	Dims	EER (%)	DCF	Params (M)	Time (ms)
ResSKNet	0	GAP	AM-softmax	512	3.29	0.3417	3.2	30.9
ResSKNet	1	GAP	AM-softmax	512	3.16	0.3253	3.2	31.3
ResSKNet	2	GAP	AM-softmax	512	2.85	0.3126	3.2	31.9
ResSKNet	3	GAP	AM-softmax	512	3.39	0.3614	3.2	32.4
ResSKNet	4	GAP	AM-softmax	512	3.68	0.3871	3.2	33.2

**Table 4 sensors-23-01203-t004:** Results of networks on Voxceleb1-O.

Model	Aggregation	Loss	Dims	EER (%)	DCF	Params (M)	Time (ms)
x-vector [7]	GAP	AM-softmax	512	4.39	0.3726	5.3	63.5
ResNet-34 [19]	GAP	AM-softmax	512	4.47	0.3909	21.4	117.6
ResNet-50 [19]	GAP	AM-softmax	512	3.89	0.3701	36.5	237.6
ResNeXt [16]	GAP	AM-softmax	512	3.85	0.3822	24.3	182.4
Res2Net [16]	GAP	AM-softmax	512	3.32	0.3442	26.0	204.2
ResSKNet	GAP	AM-softmax	512	2.85	0.3126	3.2	31.9

**Table 5 sensors-23-01203-t005:** Results of aggregation models on Voxceleb1-O.

Model	Aggregation	Loss	Dims	EER (%)	DCF	Params (M)	Time (ms)
ResSKNet	GAP [19]	AM-softmax	512	2.85	0.3126	3.2	31.9
ResSKNet	SP [7]	AM-softmax	512	2.64	0.2883	3.2	33.1
ResSKNet	SAP [21]	AM-softmax	512	2.67	0.2699	3.4	35.3
ResSKNet	ASP [23]	AM-softmax	512	2.57	0.2563	3.6	37.6
ResSKNet	NetVLAD [28]	AM-softmax	512	2.42	0.2491	3.9	40.0
ResSKNet	SSDP	AM-softmax	512	2.33	0.2298	3.4	36.3

**Table 6 sensors-23-01203-t006:** Results of different systems on the Voxceleb1-O test set.

System	Aggregation	Loss	Dims	Voxceleb1-O	
				EER (%)	DCF
ThinResNet-34 [19]	NetVLAD	AM-softmax	512	3.32	0.3391
ThinResNet-34 [19]	GhostVLAD	softmax	512	2.85	-
ResNet-34 [21]	GAP	AM-softmax	512	4.83	0.3809
ResNet-50 [21]	GAP	AM-softmax	512	3.95	0.3471
RawNet2 [41]	SP	AAM-softmax	512	2.48	0.2337
RawNet3 [41]	SP	AAM-softmax	512	2.35	0.2513
CNN+Transformer [40]	GAP	softmax	512	2.56	-
PA-ResNet50 [42]	MRMHA	AAM-softmax	512	3.32	-
ECAPA-TDNN [12]	SP	AM-softmax	128	2.38	0.2349
x-vector [7]	AP	AM-softmax	1500	3.85	0.3515
Res2Net [16]	SAP	AM-softmax	512	2.65	0.2613
ResNeXt [16]	SAP	AM-softmax	512	2.79	0.2817
Y-vector [43]	SP	AM-softmax	512	2.78	0.2694
S-vector [44]	SP	softmax	512	2.76	0.3015
ResSKNet-SSDP	SSDP	AM-softmax	512	2.33	0.2298

**Table 7 sensors-23-01203-t007:** Results of different systems on the Voxceleb1-E and Voxceleb1-H test sets.

System	Aggregation	Loss	Dims	Voxceleb1-E		Voxceleb1-H	
				EER (%)	DCF	EER (%)	DCF
ThinResNet-34 [19]	NetVLAD	AM-softmax	512	3.24	0.3471	5.57	0.4479
ResNet-34 [21]	GAP	AM-softmax	512	4.92	0.4218	7.97	0.5091
ResNet-50 [21]	GAP	AM-softmax	512	4.42	0.3822	6.71	0.4514
RawNet2 [41]	SP	AAM-softmax	512	2.87	0.2701	4.33	0.4091
x-vector [7]	AP	AM-softmax	1500	4.01	0.3868	7.33	0.4893
Res2Net [16]	SAP	AM-softmax	512	2.71	0.2653	4.42	0.3702
ResNeXt [16]	SAP	AM-softmax	512	2.74	0.2993	4.58	0.4036
Y-vector [43]	SP	AM-softmax	512	2.64	0.2701	4.33	0.3775
ResSKNet-SSDP	SSDP	AM-softmax	512	2.44	0.2559	4.10	0.3502

**Table 8 sensors-23-01203-t008:** Results of different systems on the CN-celeb test set.

Model	Aggregation	Loss	Dims	EER (%)	DCF
PA-ResNet-50 [42]	MRHMA	AM-softmax	256	13.21	0.5409
ResNet-34 [21]	SP	AM-softmax	512	16.52	0.6092
EDINet [45]	GAP	cosine loss	256	12.56	-
ResNet-50 [21]	SAP	AM-softmax	512	15.38	0.5691
ThinResNet-34 [19]	ASP	AM-softmax	512	18.37	0.6097
x-vector [7]	NetVLAD	AM-softmax	512	16.59	0.5864
Res2Net [16]	SAP	AM-softmax	512	13.72	0.5393
ResNeXt [16]	SAP	AM-softmax	512	14.55	0.5645
ResSKNet-SSDP	SSDP	AM-softmax	512	12.28	0.5051

## Data Availability

“Voxcceleb Data set” at https://www.robots.ox.ac.uk/~vgg/data/voxceleb accessed on 14 January 2023. “CN-Celeb Data set” at http://www.openslr.org/82/ accessed on 14 January 2023.

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
