# Peer review of "ResSKNet-SSDP: Effective and Light End-To-End Architecture for Speaker Recognition"

_sensors, 2023, doi:10.3390/s23031203_

Round 1

Reviewer 1 Report

In speaker recognition tasks, convolutional neural network (CNN)-based approaches have 13 shown significant success. Modeling long-term contexts and efficiently aggregating information are 14 two challenges in speaker recognition, and they have a critical impact on system performance. Pre- 15 vious research has addressed these issues by introducing deeper, wider, and more complex network 16 architectures and aggregation methods. But it is difficult to significantly improve performance with 17 these approaches because they also have trouble fully utilizing some important information. To ad- 18 dress the above issues, we propose a lighter and more efficient CNN-based end-to-end speaker 19 recognition architecture, ResSKNet-SSDP. ResSKNet-SSDP consists of a residual selective kernel 20 network (ResSKNet) and self-attentive standard deviation pooling (SSDP). ResSKNet can capture 21 long-term contexts, neighboring information, and global information, thus extracting more informa- 22 tive frame-level. I have the following major comments:

1- There are lots of basic grammar mistakes. For instance, in the abstract, it should be 20 networks, not network. Please address all of them.

2- In the introduction section, compare with following 4 papers and cited them are vital:

DOI: 10.1109/JSEN.2020.3022536

https://doi.org/10.1016/j.adhoc.2021.102520

DOI: 10.1109/UEMCON47517.2019.8992978

3- Add more results and compare them with other well-known techniques. Highlight them to see new results in the conclusion.

4- Add a comparison table consisting of benefits and comparisons with other techniques in the result.

5- The conclusion is very wide. Make is specific for the main achievement of the paper. 

Author Response

Response to Reviewer 1 Comments

Point 1: There are lots of basic grammar mistakes. For instance, in the abstract, it should be 20 networks, not network. Please address all of them.

Response 1: We had Prof. GeXiang Zhang help us check the grammar of the whole paper to make sure our grammar was correct. He is the President of the International Society for Membrane Computing, IET Fellow, and IEEE Senior Member, and has excellent English and writing skills. In addition, when using the template provided by MDPI Sensors, it displays the number of lines on the right side. Therefore, it will cause you to think that the number of rows is also part of our paper, making errors in reading, such as your proposal of 20 networks, which is actually 'ResSKNet-SSDP consists of a residual selective kernel network (ResSKNet) and self-attentive standard deviation pooling (SSDP)'.

Point 2: In the introduction section, compare with following 4 papers and cited them are vital:

DOI: 10.1109/JSEN.2020.3022536

https://doi.org/10.1016/j.adhoc.2021.102520

DOI: 10.1109/UEMCON47517.2019.8992978

Response 2: Thank you for your suggestion. We have cited the paper you have presented in the introduction section. The citation is made in the fifth paragraph of the Introduction section. We modified it in the Introduction section.

Point 3: Add more results and compare them with other well-known techniques. Highlight them to see new results in the conclusion.

Response 3: In the results section, we cite more advanced speaker recognition systems for comparison with our proposed method, such as ECAPA-TDN [1], CNN+Transformer [2], PA-ResNet-50 [3], RawNet3 [4], S-vector [5], and EDINet [6], and test them on four test sets, Voxceleb1-O, Voxceleb1-E, Voxceleb1-H, and CN-Celeb, to fully validate the effectiveness of our method. The experimental results are shown in Tables 6, 7, 8 in Section 5.2 of the Results section. We have modified it in sections 5.2 and 5.3 of the results Section. Finally, we highlight these results in our conclusion and give quantitative information. We modified it in the Results and Conclusions section.

  1. Desplanques, B.; Thienpondt, J.; Demuynck, K. ECAPA-TDNN: Emphasized Channel Attention, Propagation and Aggregation in TDNN Based Speaker Verification. Proc. Interspeech 2020, 3830-3834, doi: 10.21437/Interspeech.2020-2650.
  2. Wang, Rui et al. Multi-View Self-Attention Based Transformer for Speaker Recognition. (2021) arXiv: arXiv: 2110.05036.
  3. Wei, Y.; Du, J., Liu, H.; Wang, Q. CTFALite: Lightweight Channel-specific Temporal and Frequency Attention Mechanism for Enhancing the Speaker Embedding Extractor. Proc. Interspeech 2022, 341-345, doi: 10.21437/Interspeech.2022-10288.
  4. Jung, Jee-weon et al. Pushing the limits of raw waveform speaker recognition (2022), arXiv: arXiv:2203.08488v2.
  5. Mary, Narla John Metilda Sagaya et al. S-Vectors and TESA: Speaker Embeddings and a Speaker Authenticator Based on Transformer Encoder. IEEE/ACM Transactions on Audio, Speech, and Language Processing 30 (2020): 404-413.
  6. Li, J.; Liu, W.; Lee, T. EDITnet: A Lightweight Network for Unsupervised Domain Adaptation in Speaker Verification. Proc. Interspeech 2022, 3694-3698, doi: 10.21437/Interspeech.2022-967.

Point 4: Add a comparison table consisting of benefits and comparisons with other techniques in the result.

Response 4: We redid the tables in the results section and added a series of comparison tables consisting of benefits and comparisons with other techniques, as shown in Tables 1, 2, and 5 in Result section. Since some of the methods are not open source, we directly applied the results from the paper. We first conducted a series of ablation experiments, and the results are shown in Tables 1, 2, 3, and 4 in Result section. Among them, Table 1 shows the experimental results of our proposed ResSK conv compared with the regular convolution and SK conv on Voxceleb, which suggests that our proposed ResSK conv can better model the long-time contexts. Table 2 shows the effects of different dilation rates in ResSK conv on the system performance, and the most suitable dilation rate is selected. We can see that the larger the dilation rate is not better; as the inefficiency increases, more local information is lost, and the correlation between features becomes weaker. Table 3 shows the performance of our proposed ResSKNet compared to other state-of-the-art network structures, where we use a common training method, i.e., no attention and aggregation models are introduced. The experimental results show that our proposed ResSKNet has better performance, fewer parameters, and faster inference time and is a lighter and more efficient network structure. Table 4 shows the comparison of our proposed SSDP aggregation model with other advanced aggregation models and their impact on the system performance. The experimental results demonstrate that our proposed SSDP aggregation model can better aggregate frame-level features and obtain more differentiated utterance-level features, thus significantly improving the system performance. And then, we compared our proposed ResSKNet-SSDP speaker recognition system with other advanced speaker recognition systems on Voxceleb1-O, Voxceleb1-E, Voxceleb1-H, and CN-Celeb, and the results are shown in Tables 5, 6, and 7. In all test sets, our proposed method has a lead over the current state-of-the-art systems, which proves that our proposed ResSKNet-SSDP speaker recognition system is a lighter and more efficient speaker recognition architecture. These tables are visible in the Results section in sections 5.1, 5.2, 5.3. We modified it in the Results section.

Point 5: The conclusion is very wide. Make is specific for the main achievement of the paper.

Response 5: The previous conclusions were indeed too broad, so we have modified them for our conclusions. We first summarize our proposed methods and what problems they solve, i.e., we proposed the ResSKNet-SSDP speaker recognition system, which consists of the ResSKNe and SSDP aggregation models. The ResSKNet-SSDP network models the long-term contexts more effectively through the ResSKNet network structure. And SSDP aggregation model can capture the short- and long-term changes of frame-level features, aggregating variable-length frame-level features into fixed-length, more differentiated utterance-level features. And then, we describe quantitative information based on the experimental results confirming that our proposed method is a more efficient and lighter speaker recognition system, and more suitable for real-life use. We achieved the lowest EER/DCF of 2.33%/0.2298, 2.44%/0.2559, 4.10%/0.3502, and 12.28%/0.5051 on the noisy Voxceleb1 and CN-Celeb test sets, outperforming more than many existing methods and effectively improving the accuracy of the end-to-end speaker recognition system. Compared with the lightest x-vector, our designed ResSKNet-SSDP has 3.1M fewer parameters and 31.6ms less inference time but 35.1% better performance. Finally, we describe some limitations of the current work and describe the direction of future work. The essence of speaker recognition is a metric learning problem, and it is important that we build an efficient system to obtain more distinguishing features. It is also important to map the features to the metric space efficiently, and the key to this is the loss function. This is an area where our current work has not been done. Therefore, in future work, we will focus on improving the loss function and applying the framework we have designed to other areas. We modified it in the Conclusions section.

Reviewer 2 Report

ResSKNet-SSDP: Effective and light end-to end architecture for

speaker recognition

--------------------------------

1. The open question in the abstract is confusiong.

What is 'some important information'?

Please make it clear for readability.

2. How the proposed method overcomes the previous methods?

3. The quantitative information is required in the abstract

4. The paper needs to discuss and cite several audio representation methods such as

MFCC and handcrafted [a,b] available in the literature.

[a] https://ieeexplore.ieee.org/abstract/document/9784899

[b] https://ieeexplore.ieee.org/abstract/document/9931407

These high-quality papers explain some recent technologies for the audio representation although the domain is in biomedical sector.

5. The SOTA comparison is weak in the paper. Please compare with recent

SOTA methods for the validity.

6. The paper requires the ablative study of the proposed DL model and hyperparameters optimisation.

Author Response

Response to Reviewer 2 Comments

Point 1: The open question in the abstract is confusiong. What is 'some important information'? Please make it clear for readability.

Response 1: Here "some important information" contains three parts, which consist of global information, channel information, and time-frequency information. In the feature extractor, convolution can capture local time-frequency information, but it cannot capture essential global information due to the limitation of the convolution kernel. Since they operate in the time-frequency dimension, they ignore the channel information. In aggregation models, to aggregate variable-length frame-level features into utterance-level features, the features are usually processed using global average pooling, which only retains the channel dimension of the features, thus making the features lose important speaker information in the time-frequency dimension. Therefore, by "some important information," we mean the global and channel information that cannot be captured by the feature extractor and the time-frequency information that is lost by the aggregation model. We modified it in the Abstract section.

Point 2: How the proposed method overcomes the previous methods?

Response 2: The previous methods have three problems: firstly, convolution cannot capture global information and cannot model long-term contexts; secondly, the aggregation model loses important time-frequency information in the process of aggregating frame-level features, leading to weak discriminative utterance-level features; and thirdly, the existing speaker recognition system has large parameters and a long inference time, which does not apply to mobile devices. To solve the problem that convolution cannot capture global information, we designed the ResSK convolution, as shown in Figure 1. It performs standard and dilated convolutional layers on two parallel paths to capture local information and global information. Dilated convolution has a larger receptive field to better capture global information, and regular convolution can efficiently capture local information to better model short-term and long-term contexts. Then, we use a self-attentive module to obtain global information and adaptively adjust the weight between short-term and long-term contexts. And then, we design the SSDP aggregation model. It avoids pooling operations and uses linear attention to obtain the weights of time-frequency feature descriptors, which can select time-frequency feature descriptors more accurately. We use a class standard deviation calculation to aggregate time-frequency feature descriptors to generate utterance-level features, capturing both short-term and long-term changes in features, and thus aggregating more discriminative utterance-level features. Finally, to make our system lighter and more efficient, we designed ResSKNet and introduced SSDP into ResSKNet. We connect the output features of each stage of the ResSKNet block to make full use of the shallow and deep features. The final architecture is shown in Figure 2. ResSKNet produces more informative frame-level features and SSDP aggregates more discriminative utterance-level features, which makes our system lighter and more efficient, and thus more suitable for mobile devices.

Point 3: The quantitative information is required in the abstract

Response 3: We have modified the abstract and added quantitative information to it. We describe that we achieved the lowest EER/DCF of 2.33%/0.2298, 2.44%/0.2559, 4.10%/0.3502, and 12.28%/0.5051 in the four test sets of Voxceleb and CN-Celeb. Then, we compared it with the lightest speaker recognition system, x-vector, and the parameters of our ResSKNet-SSDP speaker recognition system decreased by 3.1M inference time decreased by 31.6ms, but the performance improved by 35.1%. These experimental results demonstrate the effectiveness of our approach and the applicability of our method to mobile devices and real life. We modified it in the Abstract section.

Point 4: The paper needs to discuss and cite several audio representation methods such as

MFCC and handcrafted [a,b] available in the literature.

[a] https://ieeexplore.ieee.org/abstract/document/9784899

[b] https://ieeexplore.ieee.org/abstract/document/9931407

These high-quality papers explain some recent technologies for the audio representation although the domain is in biomedical sector.

Response 4: Thanks to your suggestions, we have cited these papers in our paper and discussed why we chose FBank as the input feature. FBank features are more in line with the nature of the sound signal and fit the reception characteristics of the human ear. The Discrete Cosine Transform is a linear transform, which will lose some of the original highly nonlinear components of the speech signal. Before deep learning, MFCC with GMMs-HMMs was the mainstream approach for ASR, limited by algorithms. When deep learning methods came out, MFCC was not the optimal choice due to the insensitivity of neural networks to highly correlated information, and its performance in neural networks was also significantly inferior to FBank after practical verification [1]. We modified it in Section 4.2 of the Experimental Setup section.

[1] Haytham Fayek. Speech Processing for Machine Learning: Filter banks, Mel-Frequency Cepstral Coefficients (MFCCs) and What's In-Between. 2016. https://haythamfayek.com/2016/04/21/speech-processing-for-machine-learning.

Point 5: The SOTA comparison is weak in the paper. Please compare with recent

SOTA methods for the validity.

Response 5: We have selected advanced speaker identification systems from the last two years such as ECAPA-TDN [1], CNN+Transformer [2], PA-ResNet-50 [3], RawNet3 [4], S-vector [5], and EDINet [6], and test them on four test sets, Voxceleb1-O, Voxceleb1-E, Voxceleb1-H, and CN-Celeb, to fully validate the effectiveness of our method, as shown in Tables 1, 2, 3 in Section 5.2 of the Results section. We have modified it in sections 5.2 and 5.3 of the results Section. 

  1. Desplanques, B.; Thienpondt, J.; Demuynck, K. ECAPA-TDNN: Emphasized Channel Attention, Propagation and Aggregation in TDNN Based Speaker Verification. Proc. Interspeech 2020, 3830-3834, doi: 10.21437/Interspeech.2020-2650.
  2. Wang, Rui et al. Multi-View Self-Attention Based Transformer for Speaker Recognition. (2021) arXiv: arXiv: 2110.05036.
  3. Wei, Y.; Du, J., Liu, H.; Wang, Q. CTFALite: Lightweight Channel-specific Temporal and Frequency Attention Mechanism for Enhancing the Speaker Embedding Extractor. Proc. Interspeech 2022, 341-345, doi: 10.21437/Interspeech.2022-10288.
  4. Jung, Jee-weon et al. Pushing the limits of raw waveform speaker recognition (2022), arXiv: arXiv:2203.08488v2.
  5. Mary, Narla John Metilda Sagaya et al. S-Vectors and TESA: Speaker Embeddings and a Speaker Authenticator Based on Transformer Encoder. IEEE/ACM Transactions on Audio, Speech, and Language Processing 30 (2020): 404-413.
  6. Li, J.; Liu, W.; Lee, T. EDITnet: A Lightweight Network for Unsupervised Domain Adaptation in Speaker Verification. Proc. Interspeech 2022, 3694-3698, doi: 10.21437/Interspeech.2022-967.

Point 6: The paper requires the ablative study of the proposed DL model and hyperparameters optimisation.

Response 6: We add ablation experiments to the results section to validate our approach. In this study, we propose the ResSK conv, ResSKNet, and SSDP aggregation models, which together form the ResSKNet-SSDP speaker recognition system. Therefore, in the results section, we first performed a series of ablation experiments. We first validated our proposed ResSK conv by comparing it with the regular convolution and SK conv, as shown in Table 4 in Section 5.1.1. The result suggests that our proposed ResSK conv can better capture global information to model the long-time contexts. And then we tested the effect of hyperparameter dilation rate on ResSK conv and selected the most suitable dilation rate, as shown in Table 3 in Section 5.1.2. We can see that the larger the dilation rate is not better; as the inefficiency increases, more local information is lost, and the correlation between features becomes weaker. The most suitable dilation rate is 5 in this paper. Then, we compare our proposed ResSKNet with the most popular speaker recognition feature extractors to verify that ResSKNet is a more efficient and lighter architecture, as shown in Table 6 in Section 5.1.3. Finally, we compared SSDP with other aggregation models using ResSKNet as the backbone network to verify the effectiveness of SSDP and the impact of aggregation models on the performance of the speaker recognition system, as shown in Table 7 in Section 5.1.4. The experimental results demonstrate that using aggregation models can effectively improve system performance and our proposed SSDP aggregation model can better aggregate frame-level features and obtain more differentiated utterance-level features, thus significantly improving the system performance. We have modified it in sections 5.1 of the results Section.

Round 2

Reviewer 1 Report

All jobs applied. Thanks

Reviewer 2 Report

Thanks for revision. The paper is acceptable now.